# The Critical Factors Impacting Artificial Intelligence Applications Adoption in Vietnam: A Structural Equation Modeling Analysis

**Nguyen Van Phuoc**

Faculty of Business Administration, Posts and Telecommunications Institute of Technology, Ha Dong, Hanoi 12100, Vietnam; phuocnv@ptit.edu.vn

**Abstract:** The concept of artificial intelligence (AI) is the transformation of inanimate objects into intelligent beings that can reason similarly to humans. Computer systems are capable of imitating a number of human intelligence functions, such as learning, reasoning, problem solving, speech recognition, and planning. In this regard, artificial intelligence applications have been developed to assist corporations and entrepreneurs in making business decisions. Hence, the aim of the study is to investigate the adoption of AI applications at the Vietnamese organizational level. Using the core structures of the technology–organization–environment (TOE), the theoretical model was constructed based on how technical and environmental elements influence companies' technological innovation adoption decisions. Ten critical factors related to AI adoption are identified. To test the model, data were obtained from 193 senior managers who are directly in charge of information systems in both private and public companies in Vietnam. Subsequently, the Structural Equation Modeling (SEM) approach was used to analyze the data. The findings indicate that technical compatibility, relative advantage, technical complexity, technical capability, managerial capability, organizational readiness, government involvement, market uncertainty, and vendor partnership are significantly related to AI application adoption. Interestingly, the study results indicated that the relationship between organization size and AI adoption was not statistically significant. Therefore, the suggested adoption of the AI application could contribute to the existing research on the adoption of AI through the TOE. Finally, the significant government law implications and future research directions are further addressed.

**Keywords:** artificial intelligence adoption; technology organization environment model; structural equation model; Vietnam

## 1. Introduction

The development of artificial intelligence has prompted software and system engineers to devise novel methods for raising revenue, reducing expenses, and enhancing organizational effectiveness. Today, artificial intelligence (AI) is a significant competitive trend in industry Davenport and Ronanki (2018). AI is defined as "a collection of tools and technology capable of augmenting and enhancing organizational performance" Alsheibani et al. (2018). This is accomplished by creating "artificial" systems capable of resolving complex environmental difficulties, with "intelligence" referring to the emulation of human intelligence. This intelligence is essential for strategic planning and has been effectively employed by businesses to gain a competitive advantage over their competitors Varian (2018). It is widely expected that artificial intelligence (AI) will deliver benefits such as human augmentation, which should be considered while discussing economic growth Ransbotham et al. (2017). AI has been used and applied at the federal, industrial, and personal levels. Furthermore, the Government of the Socialist Republic of Vietnam set a clear policy for adopting artificial intelligence in the public sector by 2030, which is gradually gaining traction in the ASEAN region, notably the Vietnam government. Examining the significance of government bodies taking the initiative and beginning artificial

intelligence projects in their surroundings that fulfill their business needs. AI is the simulation of various human intelligence processes by computers, notably computer-related systems Agrawal et al. (2019). According to Alsheibani et al. (2018), "AI refers to both the intelligence of machines and the branch of computer science devoted to its development". While discussing the history of artificial intelligence, Alsheibani et al. (2018) describes it as the concept of transforming inanimate items into sentient beings capable of reasoning like humans. Human intelligence functions such as learning, thinking, problem solving, speech recognition, and planning are all simulated by computer systems. AI has progressed beyond robotic-like game play and knowledge representation to cognitive automation Dwivedi et al. (2021). AI is having an ever-increasing impact on corporate enterprises. AI is the top strategic technology for enterprises, according to Gartner Elliot and Andrews (2017). Google, Amazon, IBM, and Apple have all employed AI to improve consumer experiences Brynjolfsson and Mcafee (2017) and productivity Varian (2018) through easier collaboration Hunter (2018). The global adoption of AI presents a big opportunity for Vietnamese businesses Nguyen and Tran (2019). Furthermore, the survey predicts that AI and automation will help the Vietnamese economy to the tune of 1.2 trillion USD by 2030 Chua and Dobberstein (2021). Despite this excellent demonstration of AI, Soon Ghee Chua and Dobberstein (2021) study of business leaders indicated that only 6% of Vietnamese enterprises are consistently investing in AI and automation, compared to more than 25% in the US. Vietnamese businesses are increasingly lagging behind their worldwide competitors in the use of AI applications (Towards Purposeful Artificial Intelligence, 2016). According to a recent Gartner poll Elliot and Andrews (2017), the majority of businesses are still gathering data on what and how to implement AI. Many companies appear to be figuring out how to create a business case for AI deployment, as well as the organizational competencies needed to assess, build, and implement AI solutions, and are confused about the business applications of AI Ransbotham et al. (2017). As a result, in the Vietnamese context, a full understanding of AI adoption and associated drivers has yet to be produced. As a result, the purpose of this research is to acquire a complete understanding of how AI is being implemented by organizations in Vietnam.

The purpose of this study was to investigate the impact of technological context, organizational context, and environmental context on the adoption of AI applications. Data obtained from middle-level AI specialists, Information technology (IT) managers, IT executives (CIO, CEO), and IT professionals in Vietnam are used to test the study model and hypothesized linkages. Furthermore, the findings of this study contribute to empirical research on contextual factors that influence AI application adoption decisions using a large data set as opposed to a few isolated cases. Given the importance of AI application adoption in modern organizations and in the future, the findings of this study are also intended to assist AI application project managers and practitioners in formulating policies and targeting appropriate contextual factors to support effective AI application adoption.

## 2. Theoretical Background

### 2.1. Technology Adoption Perspective

Adopting new technologies is a proven strategy for corporate success Alsheibani et al. (2018). Previous research has primarily focused on innovative information technology (IT) or new system adoption at the person and organizational levels Oliveira et al. (2019). For individuals' technological acceptance practices, Alsheibani et al. (2018) theory of reasoned action (TRA) provides profound insight into how a person's conduct is influenced and led by their attitudes and norms. Ajzen advances TRA by proposing the theory of planned behavior (TPB) Ajzen (2012), which asserts that an individual's behavioral intents and behaviors are shaped by his or her attitude toward behavior, subjective norms, and perceived behavioral control. Davis (1985) offers the technology acceptance model (TAM) based on TRA to discover the factors that influence people's adoption or rejection of information technology. It implies that, when users encounter new technology, a variety of factors impact their decision regarding how and when to use it Davis (1989). Numerous

research studies have established the validity of TAM and established a link between behavioral intentions and actual system use Lu et al. (2003). TAM, on the other hand, does not account for part of an IS's qualitative elements or societal influences. Thus, Venkatesh et al. (2016) provide the unified theory of acceptance and use of technology (UTAUT) to explain users' intents to use an information system and subsequent usage behavior. Numerous studies on individual-level IT adoption examine the factors that influence an individual's decision to use a certain technology or system, such as Web 2.0 technologies. Mobile healthcare systems Lumsden and Gutierrez (2013); Mccarthy and Hayes (1981), as well as electronic banking Picoto et al. (2014). Tornatzky et al. (1990) offer the technology-organization-environment (TOE) paradigm to describe how technical and environmental elements influence companies' technological innovation adoption decisions.

As a result of the TOE, some scholars have begun to investigate the elements that influence an organization's IT adoption. For instance, Cristiano et al. (2001) conducted a survey to ascertain the prevalence of quality function deployment (QFD) in over 400 businesses in the United States and Japan. They discover that organizational variables such as motivation, managerial support, and data sources all contribute to the effectiveness of QFD implementation. Quaddus and Xu (2005) perform a qualitative field study and find four elements that influence the adoption and spread of knowledge management systems (KMS) in organizations: organizational culture, managerial support, individual advantages, and the KMS dream. Co et al. (1998) conduct an analysis of 27 management variables related to human factors affecting enterprises' adoption of modern manufacturing technologies (AMT). According to Kosaroglu and Hunt (2009), technical, leadership, managerial, and administrative capabilities all contribute to the success of new product development (NPD) projects in the telecommunications' industry. Oliveira and Martins (2008) review the research on IT adoption at the organizational level, including the TOE framework Rogers (1995); Tornatzky et al. (1990) diffusion of innovation (DOI) theory, Schalkoff (1990) institutional theory, and electronic data interchange (EDI) framework reviewed by Iacovou et al. (1995).

### 2.2. The Contexts of AI Adoption

TAM, TPB, and UTAUT have all been extensively used in research on AI adoption. They are, nonetheless, applicable to particular research. By comparison, the DOI and the TOE framework are two frequently used theories in organizational-level IT adoption research Oliveira and Martins (2011); Venkatesh et al. (2003). Rogers (1995) DOI Theory is one of the earliest social science theories. It originates in communication and is used to describe how an idea or product gains traction and spreads over time within a particular demographic or social system. Rogers defines diffusion as the process of disseminating innovation through time among social system actors Rogers (2010). According to the thesis, widespread adoption of innovation is necessary for progress and sustainability. Rogers observes that those who accept an innovation early exhibit distinct characteristics from those who acquire it later. He divides adopters into five groups: innovators, early adopters, early majority, late majority, and laggards. Additionally, when it comes to fostering innovation, methods for different groups of adopters should be distinct.

Several studies are now being conducted to evaluate the uses of artificial intelligence in specific fields Alsamhi et al. (2018); Macleish (1988); Oliveira and Martins (2011); Wang et al. (2016). Other works examine the theoretical underpinnings of AI Mitka (2012) as well as its applications Kouziokas (2017); Xu and Jia (2021). However, a few studies have been conducted on the adoption of artificial intelligence, particularly at the organizational level. For instance, Alsheibani et al. (2018) present a study framework for AI adoption, but this framework is not validated across a sample of enterprises in order to discover the elements affecting AI adoption. Additionally, their study lacks hypothesis tests and empirical validation. Due to the pervasive nature of AI and a lack of research on AI adoption at the organizational level, it is unable to directly build on current theories.

Adopting AI is a lengthy process that includes not only the procurement of software and technology but also the establishment of necessary infrastructure and resources over time. However, there is yet no empirical estimate of AI acceptance. As a result, study is required to examine the aspects that influence the proclivity of AI to adopt, as well as an organization's specific organizational competence and environmental circumstances. According to the evaluation of studies on AI adoption, the TOE framework is an excellent starting point for investigating AI adoption not only because it emphasizes the unique context in which the adoption process occurs, but also because it can be used to evaluate the elements affecting AI adoption. As a result, this study uses the TOE framework as its theoretical framework. Additionally, because scholars have combined the TOE framework and the DOI theory to evaluate IT adoption Oliveira and Martins (2011), this study takes the same method with AI adoption. As previously stated, the TOE framework is comprised of three components: the technology context, the organizational context, and the environmental context.

## 3. Research Model and Hypotheses

The analysis of the literature suggests that there is a gap in knowledge regarding the enabling variables that contribute to companies' adoption of AI, as well as how these elements interact and influence the choice to deploy AI. Hence, this study presents a research methodology based on the TOE framework and DOI theory in order to gain an insight deeper into the success variables influencing AI adoption at the organizational level. This study categorizes success variables into three categories of artificial intelligence, which is included technological context, organizational context, and external environment. As shown in Figure 1, compatibility, relative advantage, and complexity are all factors in the category of technological features of AI. The organizational context category includes the following variables: managerial support, organizational size, managerial capability, and organizational readiness. External environment factors include government involvement, market uncertainly, competitive pressure, and vendor partnerships. This section proposes a framework for the modified technology acceptance model to aid in characterizing the study's existing research problem and implementing a research model.

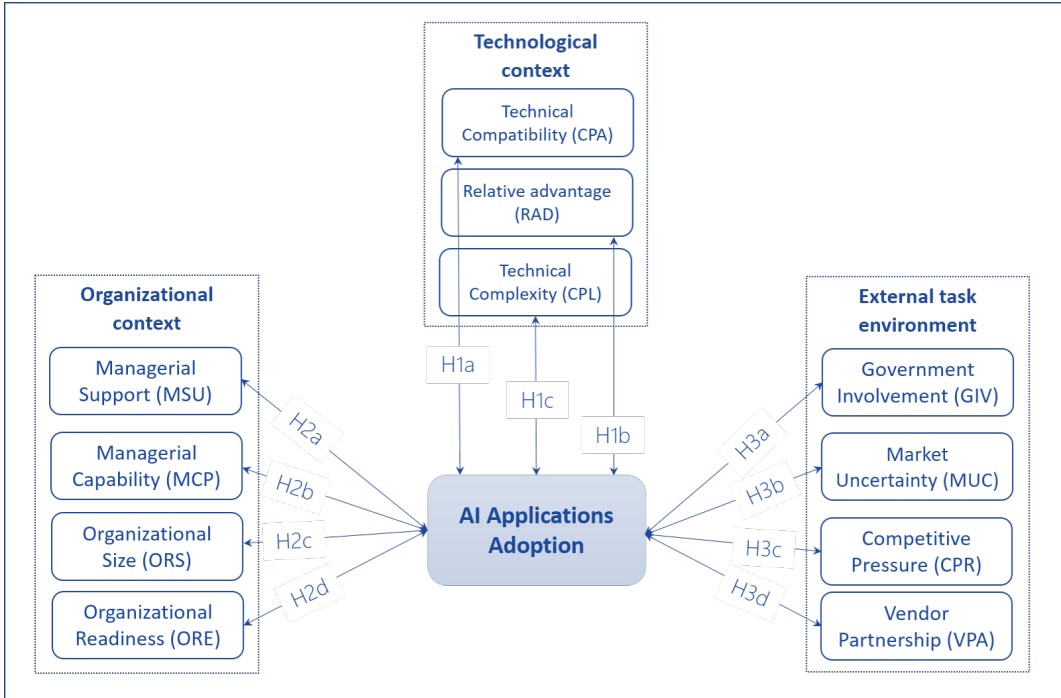

**Figure 1.** Research model.

### 3.1. Technological Context

From a technological perspective, technology demonstrates the important components in AI decision-making. Numerous studies have been conducted on the effect of innovative traits on the innovation process Chau and Tam (1997); Kwon and Zmud (1987). Although Rogers (1995) lists five qualities of innovation in DOI theory, namely compatibility, relative advantage, complexity, trialability, and observability, only the first three are reliably associated with innovation adoption at the organizational level Tornatzky et al. (1990); Wu et al. (2007).

**Technical Compatibility:** technical compatibility is a critical factor in determining whether an innovation gets adopted Azadegan and Teich (2010); Chong and Bauer (2000); Dedrick and West (2004); Oliveira et al. (2014). It refers to the extent to which an innovation and its capacity to deliver value and experience while satisfying the requirements of potential users are compatible Rogers (1995). Compatibility on a high level can result in better adoption. Artificial intelligence applications, particularly machine learning, require vast amounts of data Ding et al. (2015). If AI technology is compatible with existing IT environments, its installation is likely to be less expensive and time-consuming. As a result, AI may be more readily adopted. As a result, the following hypothesis is formed.

**H1a** *Technical compatibility is significantly and positively related to AI applications adoption.*

**Relative Advantage:** the relative advantage of an innovation is the degree to which it is seen as being superior to the strategy it substitutes Yang et al. (2013). Plessis and Smuts (2021) emphasizes that an organization's propensity to adopt new technology is influenced by the perceived advantage of innovation. As a result, new technologies that provide unambiguous benefits in terms of strategic and operational effectiveness are more likely to be accepted Greenhalgh et al. (2004). Recently, AI has been used in customer service chatbots, customer-facing speech and voice services, and automated network operation El Khatib et al. (2019). These applications help businesses cut operational expenses, improve service quality, enhance client experiences, and increase efficiency. This leads to the following hypothesis:

**H1b** *Relative advantage is significantly and positively related to AI applications adoption.*

**Technical Complexity:** the technical complexity of an innovation represents the extent to which it is considered as being comparatively difficult to comprehend and use Yang et al. (2013). The complexity of AI stems from its lack of maturity, a lack of technological competence and IT specialists, as well as its lengthy development period and high cost. Attewell (1992) observes that enterprises postpone in-house adoption of a complex technology until they have accumulated sufficient technical knowledge to successfully deploy and operate it. Currently, AI is relatively new to many businesses, which lack a good comprehension of AI applications. This leads to the following hypothesis:

**H1c** *Technical complexity is significantly and negatively related to AI applications adoption.*

### 3.2. Organizational Context

Organizational capabilities encompass the leadership, management, and managerial support resources available to facilitate an innovation's adoption. These qualities are typically organization-specific, non-transferable, and ingrained inside an organization. The resource-based view (RBV) theory can be used to determine which organizational competencies contribute to AI adoption. According to RBV, corporations gain a competitive edge by combining economically valued, difficult to copy, and nontransferable resources Garrison et al. (2015); Hannan and Mcdowell (1984).

**Managerial Support:** managerial support is crucial for any significant organizational transformation because it directs resource allocation and service integration Co et al. (1998).

Scholars have discovered that managerial support is critical for both the implementation of information systems Müller and Jugdev (2012); Nah et al. (2001); Sanders and Courtney (1985) and the acceptance of information technology Chong et al. (2009); Kim et al. (2015); Teo et al. (2006). Elbanna (2013) contends that managerial support must be consistent and continuous throughout the duration of a project's implementation, or even the project would fail. This leads to the following hypothesis:

**H2a** *Managerial support is significantly and positively related to AI applications adoption.*

**Managerial Capability:** the term "managerial capability" refers to a manager's ability to influence, motivate, and empower employees to contribute to the organization's performance and success House et al. (2002). It entails decision-making, establishing a strong workplace and culture, efficiently accomplishing goals and objectives, and cultivating creativity and innovation. In the area of information technology, managerial competency encompasses both project coordination and education & training. Firms with good management capabilities can overcome these impediments and quickly adopt new technologies. As a result, the business may quickly implement AI technologies and boost its performance, gaining a competitive edge. This leads to the following hypothesis:

**H2b** *Managerial capabilities are significantly and positively related to AI applications adoption.*

**Organization Size:** Lin and Lin (2008) the size of an organization has a significant impact on its ability to accept new innovations. Numerous studies have discovered that large organizations invest in AI more quickly and at a greater scale than other forms of investments. We propose that organization size is related to the organizational context, which has a direct impact on AI adoption. This leads to the following hypothesis:

**H2c** *Organizations size is significantly and positively related to AI applications adoption.*

**Organizational Readiness:** organizational readiness is also important when it comes to adopting AI. According to a Narrative Science survey, 59 percent of firms that are proficient in big data also use AI technologies Assael (1995). As previously mentioned, AI adoption implementations are related not only to the organization's technical readiness but also to the skill of its people resources. As a result, we believe that the availability of AI expertise, data required to train personnel in the use of AI, and technical understanding promotes the spread of AI. This leads to the following hypothesis:

**H2d** *Organizational readiness are significantly and positively related to AI applications adoption.*

*3.3. Environmental Context*

The role of institutional contexts in defining organizational structure and behaviors is emphasized by institutional theory Scott (2008). Firms are influenced by their external environment, according to Hutajulu et al. (2021). As a result, organizational decisions are influenced not just by rational efficiency goals, but also by social and cultural issues, as well as concerns about legitimacy. The external environment can both induce and dissuade enterprises from adopting new technologies. External isomorphic pressures from the government, competitors, and customers are likely to drive firms to adopt and use AI Gibbs and Kraemer (2004).

**Government Involvement:** government involvement is critical in promoting IT innovation Wang et al. (2022). The government could implement supportive strategies and policies to encourage the commercialization of new technology, as well as new rules for their development. According to Al-Hawamdeh and Alshaer (2022), the adoption of new technology is a complex process, and the framework established by the government is extremely important. This leads to the following hypothesis:

**H3a** *Government involvement is significantly and positively related to AI applications adoption.*

**Market Uncertainty:** market uncertainty factors, such as product demand, market competition, and consumer loyalty, are beyond the control of enterprises but can have an impact on their performance Hao et al. (2018). Many AI applications and applications are still in their infancy at the moment, and there is a scarcity of suitable professional and technical experts, but AI has already demonstrated significant vitality and provides enterprises with additional competitive prospects. In addition, some complicated activities, such as fingerprint identification and fact detection, are only suitable of being performed by AI programs. This leads to the following hypothesis:

**H3b** *Market uncertainty is significantly and positively related to AI applications adoption.*

**Competitive Pressure:** competitive pressure is a motivator for technological innovation. Adopting new technology is frequently a strategic necessity in order to compete in the market Dutton (2018); Lippert (2006). Firms' competitive advantages are not permanent and long-lasting, but rather transient. According to Porter and Millar (1985), IT innovation can change industry structure, change competition rules, exploit new approaches to outperform rivals, and transform the competitive environment. Firms who successfully use emerging AI applications to improve their products and services will gain a competitive advantage over their rivals. This leads to the following hypothesis:

**H3c** *Competitive pressure is significantly and positively related to AI applications adoption.*

**Vendor Partnership:** According to Assael (1995), vendor participation can considerably influence the rate of adoption and diffusion of innovative products. Vendors require a massive quantity of data to train their AI technologies, which frequently include sensitive consumer information. As a result, vendors are frequently unable to market AI solutions that are ready to use; instead, they must collaborate closely with businesses (their clients) to do AI training both during and after deployment. Partnerships between vendors can have a significant impact on the AI adoption process. As a result, AI providers can effectively sell AI applications. This leads to the following hypothesis:

**H3d** *Vendor partnership is significantly and positively related to AI applications adoption.*

## 4. Research Methodology

### 4.1. Sample and Data Collection

For empirically testing the suggested framework, we first conducted a thorough review of the literature, followed by a quantitative approach that collected data via a survey. A detailed analysis of scholarly works on technology readiness and AI was performed. Items accepted and tested in earlier studies were utilized to assist cumulative research Ahmadi et al. (2015); Cruz-Jesus et al. (2017); Lai (2017). Items assessing management hurdles and organizational preparedness factors were created specifically for this study by taking earlier research into account Picoto et al. (2014); Wright et al. (2017). To the best of our knowledge, despite the fact that the TOE has been employed in multiple IT adoptions at the company level, none of the constructs used in these studies were focused on AI adoption. As a result, a pre-test survey was conducted to check that the items were appropriate for evaluating framework dimensions in the context of this study.

In order to reach a wide number of possible participants, this study conducted a mail survey of significant Vietnamese enterprises. To verify content validity, the questionnaire items were changed based on the results of the expert interviews and polished by thorough pre-testing. Eleven constructs (Compatibility, Relative advantage, Complexity, Managerial support, Organization size, Managerial capability, Organizational readiness, Government involvement, Market uncertainty, Competitive pressure, and Vendor partnership) were operationalized as indicators of a total of 37 items. To measure these items and collect most

responses, a 7-point Likert scale ranging from "I strongly agree" (7 points) to "I strongly disagree" (1 point) is utilized. Senior managers, particularly those directly in charge of information systems in both private and public companies in Vietnam, are the intended participants. The goal was to recruit a representative sample of Vietnamese industry from a wide range of levels, backgrounds, gender and age groups, and geographical areas. The usage of the LinkedIn.com database gives advantages such as the ability to reach a big number of respondents with extremely diversified characteristics such as position, educational level, and geographical location within Vietnam, allowing the results to be more generalizable. In total, 500 invitations were sent to all industries in Vietnam. The total number of LinkedIn responses was 203, with 10 missing data points. By excluding these responses, the number of valid responses was reduced to 193, which is still suitable as a valid sample for informing the quantitative analysis Chen (2018).

### 4.2. Data Analysis

To investigate sample data and assess model fit, this study employs structural equation modeling (SEM). SEM is a technique for performing high-quality statistical analysis on multivariate data that was developed in the second generation Chin and Marcolin (1995). SEM is carried out using Analysis of Moment Structures (AMOS), a covariance-based technique for analyzing models incorporating variables with measurement error Gefen and Straub (2000). To analyze factor analysis and idea linkages, the study employs a combination of multivariate and regression analysis. SmartPLS 3.3 is used in this work to examine the measurement model and structural model. The measurement model depicts the relationships between constructs (latent variables) and their indicators (observed variables), whereas the structural model depicts the latent variables' potential causal relationships Chin et al. (2003).

## 5. Results

### 5.1. The Measurement Model

The measurement model's adequacy is determined by examining the reliability of individual items, construct validity, convergent validity, and discriminant validity of the measuring instrument. In the proposed model, eleven latent constructs (factors) and associated observable variables (indicators) are measured. Then, a factor analysis is performed to discover and corroborate the indicators under each construct about AI adoption success factors. Some signs are eliminated because their factor loadings are too low (0.4) or they are part of crossing loadings. The KMO coefficient is 0.829 (more than 0.5). The outcome of Bartlett's testing (Sig. = 0.000) suggests that the factors analysis is appropriate. From the 37 observation variables, eleven factors are extracted. The extracted variance is 72.168 percent (more than 50%). Confirmatory factor analysis (CFA) results show that all routes connecting the remaining observable variables and the constructs are significant at $p < 0.001$. According to Fornell and Larcker (1981), the percentage of extracted variance explains a model's construct validity. The overall variance explained by each indicator ranges between 50% and 80%. Table 1 displays the Cronbach's alpha (CA) value for each construct. They are all greater than 0.7, the usually accepted cutoff Kline (2013). The internal consistency of the scales is measured by composite reliability (CR). It is a more precise measure of dependability Chin and Gopal (1995). For developing appropriate model reliability, the recommended value of CR is 0.7 Gefen and Straub (2000). All of the CR values for each construct are more than the threshold. As a result, the model's build reliability is established. Convergent validity measures the consistency of many items. The Average Variance Extracted is used to calculate it (AVE). Table 2 represents the AVE values for each construct. They are all greater than the acceptable cutoff of 0.50 Fornell and Larcker (1981). This suggests that the latent constructs capture at least 50% of the measurement variation of the indicators on average Chin and Gopal (1995). Furthermore, all computed standard loadings are statistically significant at $p < 0.001$, which is greater than the allowed magnitude of 0.50 Chin and Marcolin (1995). As a result, the model's measurements show significant convergent validity.

**Table 1.** Items and descriptive statistics.

| Critical | Sub-Critical | Code | CA | R-Square | Loading |
|---|---|---|---|---|---|
| Technological context | Technical compatibility | CPA | 0.926 | | |
| | | CPA1 | | 0.753 | 0.868 *** |
| | | CPA2 | | 0.791 | 0.890 *** |
| | | CPA3 | | 0.837 | 0.915 *** |
| | | CPA4 | | 0.755 | 0.869 *** |
| | Relative advantage | RAD | 0.935 | | |
| | | RAD1 | | 0.549 | 0.741 *** |
| | | RAD2 | | 0.548 | 0.740 *** |
| | | RAD3 | | 0.695 | 0.834 *** |
| | | RAD4 | | 0.725 | 0.851 *** |
| | Technical complexity | CPL | 0.831 | | |
| | | CPL1 | | 0.744 | 0.863 *** |
| | | CPL2 | | 0.839 | 0.916 *** |
| | | CPL3 | | 0.733 | 0.856 *** |
| | | CPL4 | | 0.749 | 0.865 *** |
| Organizational context | Managerial support | MSU | 0.808 | | |
| | | MSU1 | | 0.708 | 0.841 *** |
| | | MSU2 | | 0.681 | 0.825 *** |
| | | MSU3 | | 0.716 | 0.846 *** |
| | Managerial capability | MCP | 0.911 | | |
| | | MCP1 | | 0.751 | 0.866 *** |
| | | MCP2 | | 0.812 | 0.901 *** |
| | | MCP3 | | 0.696 | 0.834 *** |
| | Organization size | ORS | 0.831 | | |
| | | ORS1 | | 0.596 | 0.702 *** |
| | | ORS2 | | 0.689 | 0.816 *** |
| | | ORS3 | | 0.682 | 0.866 *** |
| | Organizational readiness | ORE | 0.869 | | |
| | | ORE1 | | 0.695 | 0.834 *** |
| | | ORE2 | | 0.725 | 0.851 *** |
| | | ORE3 | | 0.708 | 0.841 *** |
| | | ORE4 | | 0.681 | 0.825 *** |
| External environment | Government involvement | GIV | 0.875 | | |
| | | GIV1 | | 0.716 | 0.689 *** |
| | | GIV2 | | 0.602 | 0.825 *** |
| | | GIV3 | | 0.735 | 0.789 *** |
| | Market uncertainty | MUC | 0.892 | | |
| | | MUC1 | | 0.689 | 0.737 *** |
| | | MUC2 | | 0.682 | 0.896 *** |
| | | MUC3 | | 0.593 | 0.744 *** |
| | Competitive pressure | CPR | 0.901 | | |
| | | CPR1 | | 0.786 | 0.769 *** |
| | | CPR2 | | 0.753 | 0.920 *** |
| | Vendor partnership | VPA | 0.809 | | |
| | | VPA1 | | 0.493 | 0.702 *** |
| | | VPA2 | | 0.786 | 0.887 *** |
| | | VPA3 | | 0.753 | 0.868 *** |
| | | VPA4 | | 0.787 | 0.887 *** |

Note: *** indicates significant at 1% level of significance based on t-statistics.

**Table 2.** Result of measurement model.

| Construct | Composite Reliability (CR) | Variance Inflation Factor (VIF) | Average Variance Extracted (AVE) |
|---|---|---|---|
| Technical compatibility (CPA) | 0.926 | 1.155 | 0.757 |
| Relative advantage (RAD) | 0.936 | 2.659 | 0.784 |
| Technical complexity (CPL) | 0.841 | 1.741 | 0.641 |
| Managerial support (MSU) | 0.929 | 1.275 | 0.766 |
| Managerial capability (MCP) | 0.813 | 1.546 | 0.593 |
| Organization size (ORS) | 0.901 | 2.293 | 0.752 |
| Organizational readiness (ORE) | 0.837 | 1.522 | 0.633 |
| Government involvement (GIV) | 0.871 | 2.205 | 0.629 |
| Market uncertainty (MUC) | 0.876 | 2.326 | 0.701 |
| Competitive pressure (CPR) | 0.852 | 2.490 | 0.677 |
| Vendor partnership (VPA) | 0.904 | 1.755 | 0.705 |

The Fornell–Larcker criterion is used to determine the discriminant validity of constructs, which states that the square root of AVE should be greater than the correlations between the components Fornell and Larcker (1981). As illustrated in Table 2, the square root of the AVE of each latent construct, which is bolded on the diagonal, is greater than the correlations between the latent constructs in the corresponding columns and rows. As a result, the constructs' discriminant validity is established. Additionally, the inter-item correlations are all less than 0.90 Bagozzi et al. (1991), demonstrating that each concept is different. While certain constructs have a marginally low level of construct validity, the majority of constructions have adequate levels of validity and reliability. Thus, the model's constructs' validity and reliability are established.

Multicollinearity occurs when there is a significant degree of correlation between predictor variables, resulting in unreliable and unstable regression coefficient estimations. The variance inflation factor (VIF), which is defined as the amount by which the standard error increases due to collinearity, is used to diagnose multicollinearity. Examining the correlation table for indications of multicollinearity amongst the eleven latent variables (Table 2) reveals that all VIF with latent variable scores are less than the threshold value of 5.0 Tabri and Elliott (2012). VIF values vary from 1.155 to 2.659, as shown in (Table 3) This indicates that the predictor variables are not multicollinear.

**Table 3.** Latent variable correlations.

| Construct | 1 | 2 | 3 | 4 | 5 | 6 | 7 | 8 | 9 | 10 | 11 |
|---|---|---|---|---|---|---|---|---|---|---|---|
| CPA | **0.883** | | | | | | | | | | |
| RAD | 0.720 ** | **0.895** | | | | | | | | | |
| CPL | 0.311 ** | 0.323 ** | **0.823** | | | | | | | | |
| MSU | −0.441 ** | −0.329 ** | −0.108 | **0.866** | | | | | | | |
| MCP | 0.192 ** | 0.289 ** | 0.442 ** | 0.018 | **0.791** | | | | | | |
| ORS | 0.512 ** | 0.571 ** | 0.535 ** | −0.211 ** | 0.379 ** | **0.876** | | | | | |
| ORE | 0.259 ** | 0.258 ** | 0.545 ** | −0.07 | 0.463 ** | 0.411 ** | **0.785** | | | | |
| GIV | 0.541 ** | 0.590 ** | 0.504 ** | −0.144 * | 0.404 ** | 0.591 ** | 0.443 ** | **0.802** | | | |
| MUC | 0.576 ** | 0.669 ** | 0.390 ** | −0.281 ** | 0.270 ** | 0.585 ** | 0.296 ** | 0.544 ** | **0.835** | | |
| CPR | 0.644 ** | 0.688 ** | 0.312 ** | −0.264 ** | 0.283 ** | 0.547 ** | 0.286 ** | 0.519 ** | 0.654 ** | **0.819** | |
| VPA | 0.554 ** | 0.568 ** | 0.240 ** | −0.285 ** | 0.208 ** | 0.476 ** | 0.213 ** | 0.379 ** | 0.501 ** | 0.601 ** | **0.839** |

Note: Bold numbers on the diagonal are the square root of the AVE. ** Correlation is significant at the 0.01 level (2-tailed). * Correlation is significant at the 0.05 level (2-tailed).

*5.2. Assessing the Structural Model and Hypotheses Testing*

As validate the hypothesized relationships, an examination of the structural model was conducted. The structural model was then evaluated using the structural equation model SEM-PLS. The path coefficients, coefficient of determination, and predictive significance of the measurement model were evaluated in order to examine and validate it. The method

of path coefficients encapsulates the relationships between the constructions. As shown in Table 4, 11 hypotheses (H1a, H1b, H1c, H2a, H2b, H2c, H2d, H3a, H3b, H3c, and H3d) have significant paths leading to the endogenous variable, whereas two hypotheses (H2c, and H3c) are rejected (path coefficients < 0.20). $R^2$ is the coefficient of determination and f2 denotes the size of variation explained by all external constructs in the endogenous construct. In Leguina (2015), if the results of a value greater than 0.670 are considered "substantial", 0.330 are considered "moderate", and 0.190 are considered "weak". Our findings indicate that the $R^2$ value is 0.872, which indicates a high level of prediction accuracy. Leguina (2015) finally defines f2 values greater than 0.35 as "high", values between 0.15 and 0.35 as "medium", values between 0.02 and 0.15 as "low", and values less than 0.02 as "weak". Our research found that the f2 of (CPA, ORE, MCP, and MSU) is high and the f2 of OS is weak (less than 0.02); however, the f2 of CPR and GIV is less than 0.02.

**Table 4.** Hypothesis test results.

| Hypothesis Paths | | Standard Path Coefficient ($\beta$) | *p*-Value | Results |
| --- | --- | --- | --- | --- |
| H1a | Technical compatibility —> AI adoption | 0.803 | *** | Support |
| H1b | Relative advantages —> AI adoption | 0.157 | 0.019 ** | Support |
| H1c | Complexity —> AI adoption | −0.223 | *** | Support |
| H2a | Managerial support —> AI adoption | 0.206 | 0.011 ** | Support |
| H2b | Managerial capability —> AI adoption | 0.416 | *** | Support |
| H2c | Organizational size —> AI adoption | −0.028 | 0.703 | Not support |
| H2d | Organizational readiness —> AI adoption | 0.758 | *** | Support |
| H3a | Government involvement —> AI adoption | −0.304 | *** | Support |
| H3b | Market uncertainty —> AI adoption | 0.149 | 0.047 ** | Support |
| H3c | Competitive pressures —> AI adoption | 0.036 | 0.519 | Not support |
| H3d | Vendor partnerships —> AI adoption | 0.113 | 0.048 ** | Support |

Note: *** and ** indicates significant at 1% and 5% level of significance based on t-statistics.

## 6. Discussion

The main aim of the study was to identify the critical factors affecting artificial intelligence application to adopt in Vietnam. To achieve the research objective, the study focused on the TOE framework and DOI theory in order to gain an insight deeper into the success variables influencing AI adoption at the organizational level. The study employed Structural Equation Modeling is applied to analyze the data. The results indicate that managerial capability is significantly related to innovation attributes of AI. Stronger managerial capability creates a better IT environment for AI adoption and reduces the difficulty of applying AI technologies. These results suggest that organizational size and competitive pressure do not play a role in the process of AI adoption, but government involvement and vendor partnership are critical factors for AI adoption. This means that good vendors and supplier partnerships can help firms adopt AI and government involvement can influence AI adoption. However, there is no positive relationship between AI adoption and market uncertainty and competitive pressure, respectively.

Given that conceptual framework for AI acceptance is still in the early stages, one goal of this study was to investigate AI adoption from an organizational standpoint. In terms of organizational environment, the data show that managerial support is one of the most powerful predictors of AI adoption. The findings of this study are consistent with those of Leach (2021); Zhu and Kraemer (2005), who found that managerial assistance had a considerable beneficial impact on new technology adoption. Furthermore, our findings provide additional evidence of the importance that individuals play in AI adoption. The importance of organizational preparedness implies that technological capabilities such as technology infrastructure, data structure, and human capital are crucial in determining whether or not a business adopts AI. According to the findings, organizations with a higher level of preparation tend to achieve a higher level of AI adoption. Hence, one of the characteristics of AI adopters is the attempt to develop hybrid capable abilities to support Artificial Intelligence technologies. In the instance of Vietnamese organizations, this may

be explained by implying that they may have held sufficient related expertise to overcome AI obstacles.

Surprisingly, this study discovered that the relationship between organization size and AI adoption was not statistically significant. These findings contradict those of Walczak (2018), who discovered that organization size had a favorable effect on AI and the adoption of innovative innovations. This could be explained by the rise of smaller technology-inspired start-ups. Furthermore, large organizations may be hampered by structural inertia as a result of having several levels of bureaucracy. According to the findings of this study, AI adoption is not a phenomenon dominated by large organizations. Our findings show that using organization size as a significant factor to better understand AI adoption is insufficient. This could be explained by the rise of smaller technology-inspired start-ups.

## 7. Conclusions

This study is an early investigation of AI applications adoption at the organizational level, incorporating well-established theories into a novel innovation. Our research provides a foundation for future research on why and how organizations use AI. It can be used as a starting point for further study on AI adoption in various industries. This contribution demonstrated the importance of offering guidance and tools for investigating the topic of AI adoption. The levels of abstract idea provide an overview of potential study topics. Our research makes significant contributions from both theoretical and practical perspectives, as well as offering up exciting future research options. The current study provides different insights into the underlying components that explain the AI-specific aspects that influence an organization's intention to adopt AI. This contribution begins with a definition of AI from an IS and organizational standpoint. Furthermore, this study adds to the current body of knowledge about technology adoption. To give an extended framework, this study blends known theories and in-depth research literature in AI. As the literature study demonstrated, little research has been conducted to identify what factors lead enterprises to adopt AI. As a result, this study supports the organizational context and innovative features that influence AI adoption. The findings confirm that IS theories (TOE and DOI) as a theoretical underpinning, as embedded in the AI adoption framework, can provide a more comprehensive understanding of successful AI adoption at the organizational level, but it has some limitations. First, the study was conducted using data from Vietnamese managers, which is a tiny sample size compared to the overall firm's AI application adoption. Therefore, the other elements such as the impact of government laws on AI adoption may be the subject of future research. Subsequent research should examine these issues and expand on the findings of this exploratory research to better understand the acceptance of AI and their real application in Vietnam and other scenarios.

**Funding:** This research was funded by the Posts and Telecommunications Institute of Technology.

**Institutional Review Board Statement:** Not applicable.

**Informed Consent Statement:** Not applicable.

**Data Availability Statement:** Not applicable.

**Acknowledgments:** The author wishes to express their gratitude to the Posts and Telecommunications Institute of Technology, Vietnam, for financial support for this research.

**Conflicts of Interest:** The author declares no conflict of interest.

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
