# Peer review of "The Critical Factors Impacting Artificial Intelligence Applications Adoption in Vietnam: A Structural Equation Modeling Analysis"

_economies, doi:10.3390/economies10060129_

Round 1
Reviewer 1 Report
Since AI is changing very fast, in Introduction part, more recent literature would be appreciated. The same applies for Theoretical Background part. References are missing from the last five years…
Moreover, Introduction should be more affirmative. No need to present question in than detail.
Research Model and Hypotheses are well stated and clearly presented.
Research Methodology is well structured and the approach is clearly presented. The sample is rather sufficient and the approach of data collection along with the validity tests are pretty good.
Structural Equation Modeling is applied to analyze the data. Good, very good method. AMOS search in depth the hypotheses and produces valid results.
Hypotheses and results are clearly explored and presented.
Discussion is incorporated to Results part and this is good.
Concluding part should be enhanced by presenting limitations and more precisely managerial implications.
Finally, a more thorough review on the language would be appreciated. More academic approach would enhance the quality of the paper.
Overall, the paper is rather well established and just minor amendments are needed.
Author Response
Dear Reviewer:
I wish to submit the revised manuscript titled “Critical Factors Affecting Artificial Intelligence Application Adoption in Vietnam.”
We thank you and the reviewers for your thoughtful suggestions and insights.
The manuscript has been rechecked and the necessary changes have been made in accordance with the reviewers’ suggestions. The responses to all comments have been prepared and attached herewith/given below.
Thank you for your consideration. I look forward to hearing from you.
Sincerely,
The Author

Reviewer 2 Report
The paper asseses technological, organizational and environmental factors on decisions about AI applications. It presents data from managers and engineers in Vietnam to test the hypothesized model.
The research content is significant, scientifically sound, and interesting to readers.
The scientific method is well applied until data collection. But then, the data is stated without interpretation.
The article needs a section "6. Discussion" that explains what the data means, but without speculation.
Also the conclusion now is only a summary. Here, instead, the authors should state answers to the initial research questions / hypotheses that are backed by the data.
Moreover, the paper would profit from some data visualization.
Finally, I found some minor errors:
P. 4, line 128: &
Line 170: "Rogers" instead of "E" is the last name.
Line 300: Research model is missing.
Author Response

(The authors gave the same response as above.)
